# Role of D-aminoacyl-tRNA deacylase beyond chiral proofreading as a cellular defense against glycine mischarging by AlaRS

Komal Ishwar Pawar, Katta Suma, Ayshwarya Seenivasan, Santosh Kumar Kuncha, Satya Brata Routh, Shobha P Kruparani, Rajan Sankaranarayanan*

CSIR–Centre for Cellular and Molecular Biology, Hyderabad, India

**Abstract** Strict L-chiral rejection through Gly-*cis*Pro motif during chiral proofreading underlies the inability of D-aminoacyl-tRNA deacylase (DTD) to discriminate between D-amino acids and achiral glycine. The consequent Gly-tRNA$^{Gly}$ 'misediting paradox' is resolved by EF-Tu in the cell. Here, we show that DTD's active site architecture can efficiently edit mischarged Gly-tRNA$^{Ala}$ species four orders of magnitude more efficiently than even AlaRS, the only ubiquitous cellular checkpoint known for clearing the error. Also, DTD knockout in AlaRS editing-defective background causes pronounced toxicity in *Escherichia coli* even at low-glycine levels which is alleviated by alanine supplementation. We further demonstrate that DTD positively selects the universally invariant tRNA$^{Ala}$-specific G3•U70. Moreover, DTD's activity on non-cognate Gly-tRNA$^{Ala}$ is conserved across all bacteria and eukaryotes, suggesting DTD's key cellular role as a glycine deacylator. Our study thus reveals a hitherto unknown function of DTD in cracking the universal mechanistic dilemma encountered by AlaRS, and its physiological importance.

*For correspondence: sankar@ccmb.res.in

## Introduction

D-aminoacyl-tRNA deacylase (DTD) is a key factor that keeps chiral errors away from the translational machinery by allowing only L-amino acids to form proteins and has therefore been implicated in perpetuation of homochirality in the protein world (*Calendar and Berg, 1967*; *Soutourina et al., 1999*, *2000*). The design principle by which this remarkable configurational specificity is achieved by DTD involves only strict L-chiral rejection, rather than D-chiral selection. An invariant cross-subunit Gly-*cis*Pro motif forms the structural and mechanistic basis for DTD's enantioselection (*Ahmad et al., 2013*). Thus, the architecture of DTD's chiral proofreading site is such that it cannot prevent misediting of achiral glycine charged on tRNA$^{Gly}$ and seems to have an inherent flaw. The glycine 'misediting paradox' is, however, effectively resolved through protection of the cognate achiral substrate by elongation factor thermo unstable (EF-Tu) (*Routh et al., 2016*).

While occasional chiral errors that occur during aminoacylation are cleared by DTD to ensure the accuracy of aminoacyl-tRNAs present in the cellular pool, a major role is played by editing functions associated with about half of the 20 aminoacyl-tRNA synthetases (aaRSs) to rectify the incorrect pairing of a similar non-cognate L-amino acid with a tRNA (*Guo and Schimmel, 2012*; *Ibba and Soll, 2000*). These aaRSs can edit the non-cognate amino acid at the aminoacylation site itself after the amino acid has been activated (i.e. formation of aminoacyl-AMP using ATP as a substrate) but prior to its transfer to the tRNA (pre-transfer editing). Alternatively, proofreading can happen at a distinct editing site after the activated non-cognate amino acid has been esterified with the tRNA (post-

**eLife digest** Proteins are made up of many different building blocks called amino acids, which are linked together in chains. The exact order of amino acids in a protein chain is important for the protein to work properly. When a cell makes proteins, molecules known as transfer ribonucleic acids (or tRNAs for short) bind to specific amino acids to guide them to the growing protein chains in the correct order.

Most amino acids – except one called glycine – have two forms that are mirror images of one another, known as left-handed (L-amino acids) and right-handed (D-amino acids). However, only L-amino acids and glycine are used to make proteins. This is because of the presence of multiple quality control checkpoints in the cell that prevent D-amino acids from being involved. One such checkpoint is an enzyme called D-amino acid deacylase (DTD), which removes D-amino acids that are attached to tRNAs.

Other enzymes are responsible for linking a particular amino acid to its correct tRNA. Along with mistaking D-amino acids for L-amino acids, these enzymes can also make errors when they have to distinguish between amino acids that are similar in shape and size. For example, the enzyme that attaches L-alanine to its tRNA can also mistakenly attach larger L-serine or smaller glycine to it instead. Previous research has shown that attaching L-serine to this tRNA can lead to neurodegeneration in mice, whereas attaching glycine does not seem to cause any harm. It is not clear why this is the case.

Pawar et al. investigated how incorrectly attaching glycine or L-serine to the tRNA that usually binds to L-alanine affects a bacterium called *Escherichia coli*. The experiments show that, if the mistake is not corrected, glycine can be just as harmful to the cells as L-serine. The reason that glycine appears to be less of a problem is that the DTD enzyme is able to remove glycine, but not L-serine, from the tRNA. Further experiments show that DTD can play a similar role in a variety of organisms from bacteria to mammals.

The findings of Pawar et al. extend the role of DTD beyond preventing D-amino acids from being incorporated into proteins. The next step is to understand the role of this enzyme in humans and other multicellular organisms, especially in the context of nerve cells, where it is present at high levels.

transfer editing). These proofreading processes are so crucial that even mild defects can lead to adverse cellular outcomes like cell growth retardation, neurodegeneration, cardiomyopathy and cell death (*Bacher et al., 2005*; *Bullwinkle et al., 2014*; *Karkhanis et al., 2007*; *Korencic et al., 2004*; *Lee et al., 2006*; *Liu et al., 2014*; *Lu et al., 2014*; *Moghal et al., 2016*; *Nangle et al., 2002*; *Roy et al., 2004*), although a compromise in editing can also be beneficial as it helps the organism to tide over stress conditions (*Moghal et al., 2014*).

Being comparatively small and similar to the cognate alanine, glycine and serine are misactivated by alanyl-tRNA synthetase (AlaRS) at significantly high frequencies of 1/240 and 1/500, respectively, relative to alanine (*Tsui and Fersht, 1981*) (*Figure 1*). However, these misactivation rates are much higher than the overall error rates of ~$10^{-4}$–$10^{-3}$ observed during protein biosynthesis (*Ogle and Ramakrishnan, 2005*). Once (mis)activated, these non-cognate amino acids are mischarged on tRNA$^{Ala}$ by AlaRS. This creates a unique mechanistic challenge for the editing domain of AlaRS, which has to specifically remove two non-cognate amino acids—the larger serine and the smaller glycine—attached to tRNA$^{Ala}$ without acting on the cognate alanine, which is intermediate in size between serine and glycine. This 3.5-billion-year-old double-discrimination problem is shown to be unavoidable for AlaRS in all forms of life (*Guo et al., 2009*). It has also been shown that serine mischarging on tRNA$^{Ala}$ is detrimental to the cell and even a mild deficiency in the proofreading activity of AlaRS leads to cell death and severe neuropathologies in mouse (*Lee et al., 2006*; *Liu et al., 2014*). The problem is so severe that several standalone *trans*-editing modules (collectively called AlaXs), which are homologous to AlaRS *cis*-editing domain, have come into being. However, these *trans*-editing factors are not ubiquitously present; their distribution is more in archaea than in eukaryotes and bacteria (*Guo and Schimmel, 2012*). Surprisingly, only archaeal AlaXs are known to clear

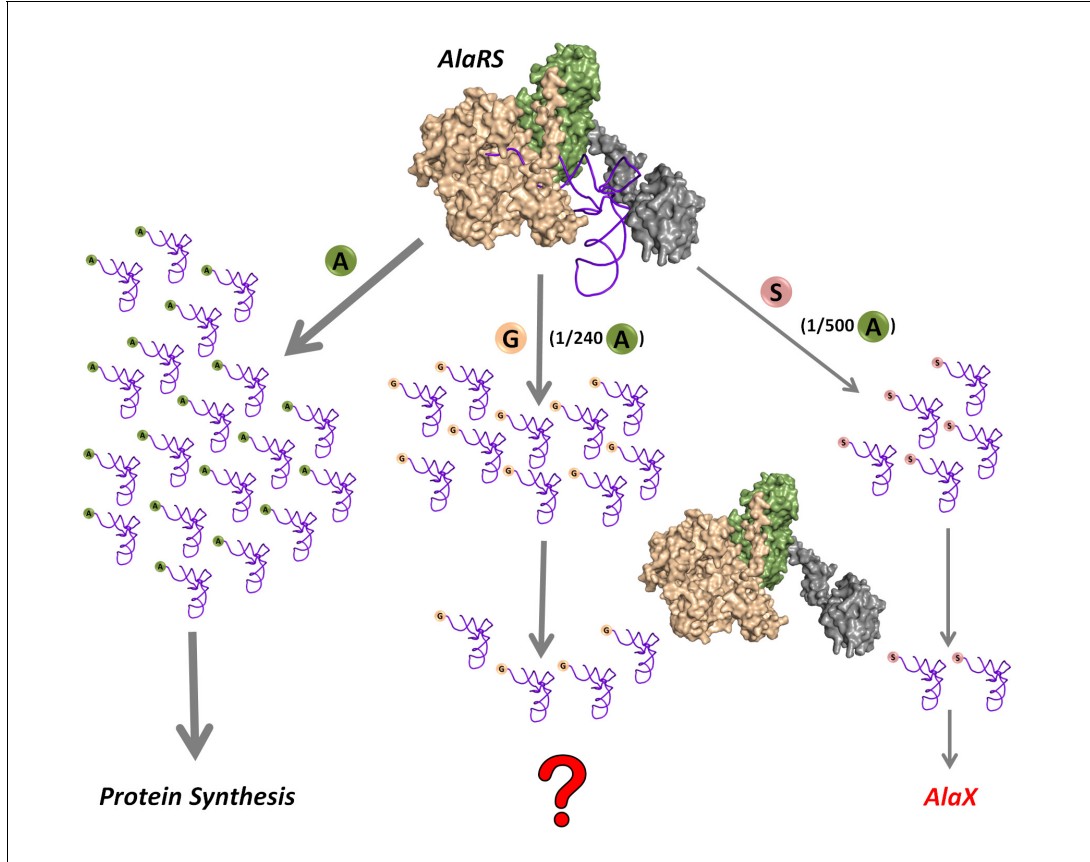

**Figure 1.** Mischarging by AlaRS. AlaRS activates and charges alanine (**A**) to form cognate Ala-tRNA$^{Ala}$ which is routed for protein synthesis. In this process, AlaRS also misactivates glycine (G) and serine (S) at frequencies of 1 per 240 alanine and 1 per 500 alanine, respectively (**Tsui and Fersht, 1981**). The two non-cognate amino acids are then charged on tRNA$^{Ala}$ to produce Gly-tRNA$^{Ala}$ and Ser-tRNA$^{Ala}$ species, with glycine mischarging being nearly twice that of serine. Since AlaRS does not distinguish much between Gly-tRNA$^{Ala}$ and Ser-tRNA$^{Ala}$ while clearing the two, higher levels of Gly-tRNA$^{Ala}$ might accumulate in the cell. However, there are additional free-standing *trans*-editing factors called AlaX (found in all domains of life but not in all organisms), which are known to edit mainly Ser-tRNA$^{Ala}$. This leads to a fundamental question as to how the problem of Gly-tRNA$^{Ala}$ editing is solved in the cellular context.

both Ser-tRNA$^{Ala}$ and Gly-tRNA$^{Ala}$ (**Ahel et al., 2003**); eukaryotic AlaXs have been shown to act as cellular redundancies to edit only Ser-tRNA$^{Ala}$ (**Guo et al., 2009**), whereas biochemical activity of bacterial AlaXs is yet to be probed (**Figure 1**). These findings corroborated the notion that only serine mischarging by AlaRS poses the major threat to the cell (**Guo et al., 2009**; **Lee et al., 2006**; **Liu et al., 2014**).

In the current study, we show that there is significant glycine mischarging by AlaRS in the presence of EF-Tu which can be equally pernicious as serine mischarging. We demonstrate that DTD plays an active and crucial role in preventing the accumulation of mischarged Gly-tRNA$^{Ala}$ species. A cell lacking DTD in AlaRS editing-defective background displays pronounced toxicity toward even low levels of glycine which is, nevertheless, alleviated by alanine supplementation. Our data also indicate that DTD has selectivity for the G3•U70 wobble base pair that is unique to tRNA$^{Ala}$, suggesting that in the primordial scenario, DTD could have been recruited primarily as a glycine-removing factor. Our study thus brings to the fore three important aspects of translational fidelity, which were underappreciated or unknown so far. Firstly, glycine, like serine, can be toxic and deleterious to the cell under conditions wherein the cell is deficient in disposing of the mischarged Gly-tRNA$^{Ala}$ species. Secondly, how the design of the active site of DTD, notwithstanding its unwarranted activity on Gly-tRNA$^{Gly}$, is used to efficiently decouple glycine mischarged on tRNA$^{Ala}$ despite the presence of EF-Tu, thereby fortifying translational fidelity. Thirdly, there is a positive selection of the element

(s) of tRNA[Ala] by DTD, indicating for the first time the role of tRNA elements in modulating DTD's activity.

## Results

### Mischarging by AlaRS leads to significant accumulation of Gly-tRNA[Ala]

To test whether Gly-tRNA[Ala] is accumulated due to mischarging of glycine on tRNA[Ala] by AlaRS, we performed aminoacylation assays in the presence of EF-Tu. In comparison to alanine charging, significant glycine mischarging was observed. Furthermore, the level of glycine mischarging was about twice that of serine mischarging (*Figure 2a*). This clearly indicated that even with full AlaRS editing potential, there can be significant accumulation of Gly-tRNA[Ala] species in the cell. Moreover, this is in accordance with the twofold higher misactivation rate of glycine by AlaRS when compared to that of serine (*Tsui and Fersht, 1981*). We then checked the accumulation of Gly-tRNA[Ala] when AlaRS editing was compromised, and it was found to be significantly high, almost equal to the level of Ala-tRNA[Ala] formation (*Figure 2—figure supplement 1*). To accomplish a compromise in the proofreading activity of AlaRS, a known editing site mutation (viz., C666A) in AlaRS from *Escherichia coli* was used (*Beebe et al., 2003*). The above data lead to a couple of fundamental questions: (a) why does a defect in the same editing domain that edits both serine and glycine from tRNA[Ala] cause toxicity only due to serine? and (b) how does the cell tackle the problem of glycine mischarging? Furthermore, based on structural considerations and evolutionary substitution patterns of alanine, where it is replaced more by serine than by glycine (*Betts et al., 2003*), it is to be expected that substitution of glycine for alanine is more detrimental than substitution of serine.

### DTD effectively decouples glycine mischarged on tRNA[Ala]

The leads for the solution to this puzzle came when, surprisingly, we found that the activity of DTD on Gly-tRNA[Ala] was ~1000-fold more than that on Gly-tRNA[Gly] (as discussed later). Moreover, although the ratio of activated EF-Tu to DTD in our assays (*viz.*, ~200 nM to 5 or 10 pM) was much higher than the cellular ratio (*viz.*, ~200:1) (*Li et al., 2014*), DTD could easily edit Gly-tRNA[Ala] in the presence of EF-Tu (*Figure 2b*). This was unlike the case of Gly-tRNA[Gly], wherein EF-Tu showed significant protection of the cognate achiral substrate from DTD (*Routh et al., 2016*). The deacylation of Gly-tRNA[Ala] by DTD was so striking that ~20,000 times more AlaRS (which is the only universally occurring editing factor for Gly-tRNA[Ala]) as compared to DTD was required for similar kind of deacylation under identical conditions (*Figures 2b* and *3b*). In addition, unlike DTD, AlaRS showed a significant decrease in deacylation activity on Gly-tRNA[Ala] when tested in the presence of EF-Tu (*Figure 3a,b*). Moreover, our assays demonstrated that DTD is not only efficient in eliminating Gly-tRNA[Ala] despite the presence of EF-Tu (*Figure 2b*) but can also very effectively prevent the accumulation of Gly-tRNA[Ala] during aminoacylation by AlaRS in the presence of EF-Tu (*Figure 2a*).

### DTD has significantly higher activity than AlaRS for clearing mischarged Gly-tRNA[Ala]

Considering the high activity of DTD on Gly-tRNA[Ala], we probed the relative efficiencies of DTD and AlaRS in editing Gly-tRNA[Ala]. To this end, we performed competition assays involving AlaRS, DTD and EF-Tu. In both aminoacylation and deacylation conditions, we found that DTD deacylated Gly-tRNA[Ala] even in the presence of EF-Tu and AlaRS at just 10 pM concentration of DTD (*Figures 2a* and *3c*). Considering the cellular ratio of DTD to AlaRS (viz, ~1:5) (*Li et al., 2014*) and their relative activities on Gly-tRNA[Ala], it is evident that when DTD is present, it will eliminate Gly-tRNA[Ala] more efficiently than AlaRS if the non-cognate achiral substrate is released in solution from the synthetase. In this context, it is important to note that compared to Class I synthetases, enzymes belonging to Class II, which include AlaRS, have been shown to have faster product release rates (*Zhang et al., 2006*). Hence, Class II aaRSs require resampling of the released mischarged product to edit the cytosolic pool of mischarged tRNAs (*Ling et al., 2009*). This makes even more sense as regards AlaRS, since our structural analysis of AlaRS in complex with tRNA[Ala] (PDB id: 3WQY) suggests that it would be very difficult for the CCA-arm at the 3' end of tRNA[Ala] harboring the non-cognate amino acid to flip from the aminoacylation site to the editing site without undergoing major conformational changes (*Naganuma et al., 2014*). Such a dynamics would naturally facilitate a faster release of the

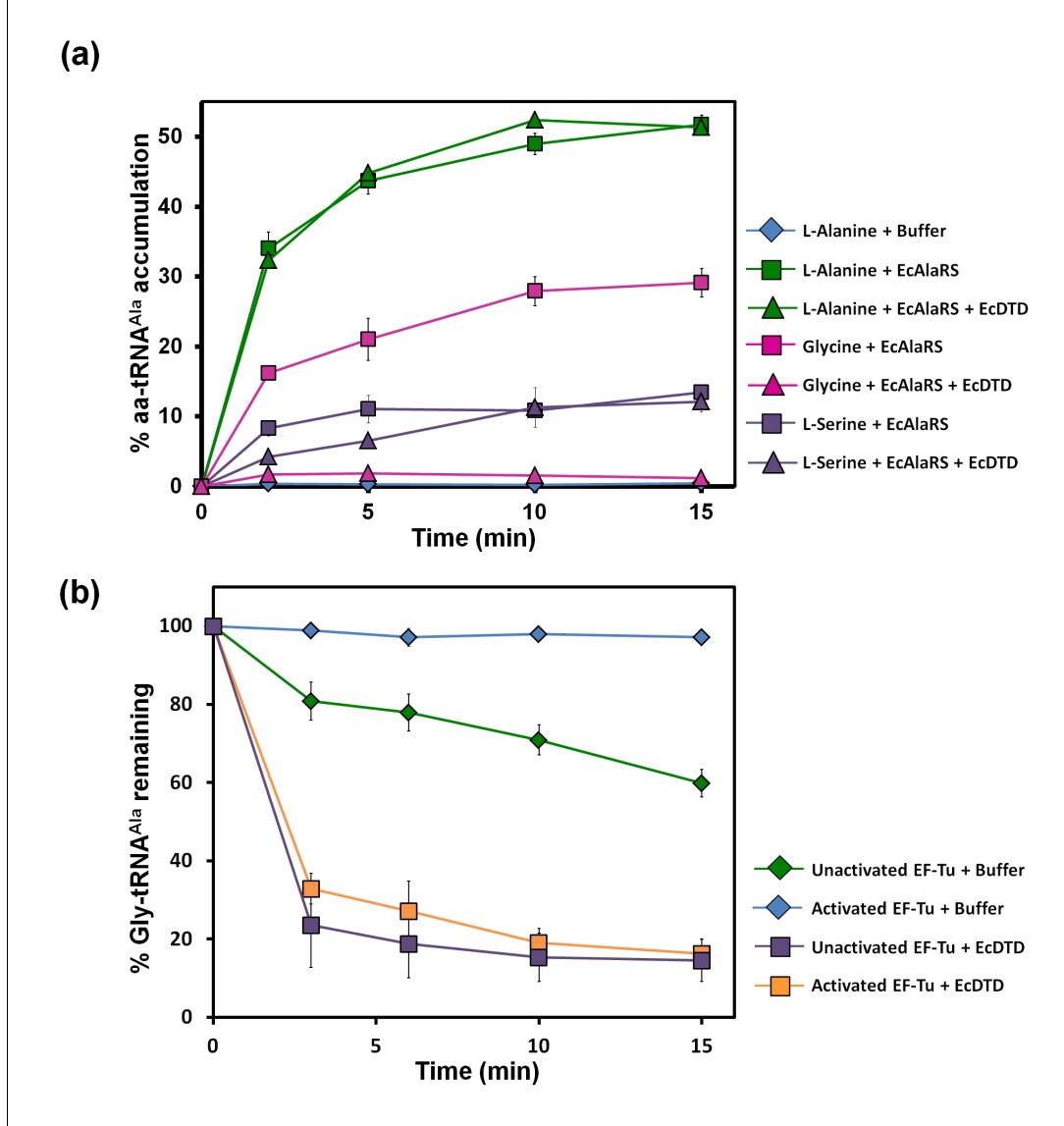

**Figure 2.** Misacylation of tRNA$^{Ala}$ with glycine by AlaRS and its prevention/rectification by DTD. (**a**) Aminoacylation of tRNA$^{Ala}$ by EcAlaRS in the presence of activated EF-Tu: L-alanine (green square), L-alanine and 10 pM EcDTD (green triangle), glycine (pink square), glycine and 10 pM EcDTD (pink triangle), L-serine (purple square), L-serine and 10 pM EcDTD (purple triangle). No enzyme control (blue diamonds) reaction had all the components of the reaction (with L-alanine) except for EcAlaRS. (**b**) Deacylation of Gly-tRNA$^{Ala}$ in the presence of unactivated EF-Tu (green diamond), activated EF-Tu (blue diamond), 5 pM EcDTD and unactivated EF-Tu (purple square), 5 pM EcDTD and activated EF-Tu (orange square). Error bars indicate one standard deviation from the mean of triplicate readings.

The following source data and figure supplement are available for figure 2:

**Source data 1.** Misacylation of tRNA$^{Ala}$ and deacylation of Gly-tRNA$^{Ala}$ in the presence of EF-Tu.

**Figure supplement 1.** Accumulation of Ala/Gly/Ser-tRNA$^{Ala}$ during aminoacylation by EcAlaRS C666A in the presence of EF-Tu.

misacylated product in solution, implying that a significant fraction of Gly-tRNA$^{Ala}$ and Ser-tRNA$^{Ala}$ is released from AlaRS prior to their recapture for proofreading by the *cis*-editing domain. Moreover, our own data, in which we observed significant accumulation of Gly-tRNA$^{Ala}$ in the presence of EF-Tu (***Figure 2a***), corroborate the aforementioned aspects of Class II synthetases in general and AlaRS in particular. Overall, the above data suggest that the problem of glycine mischarging by AlaRS

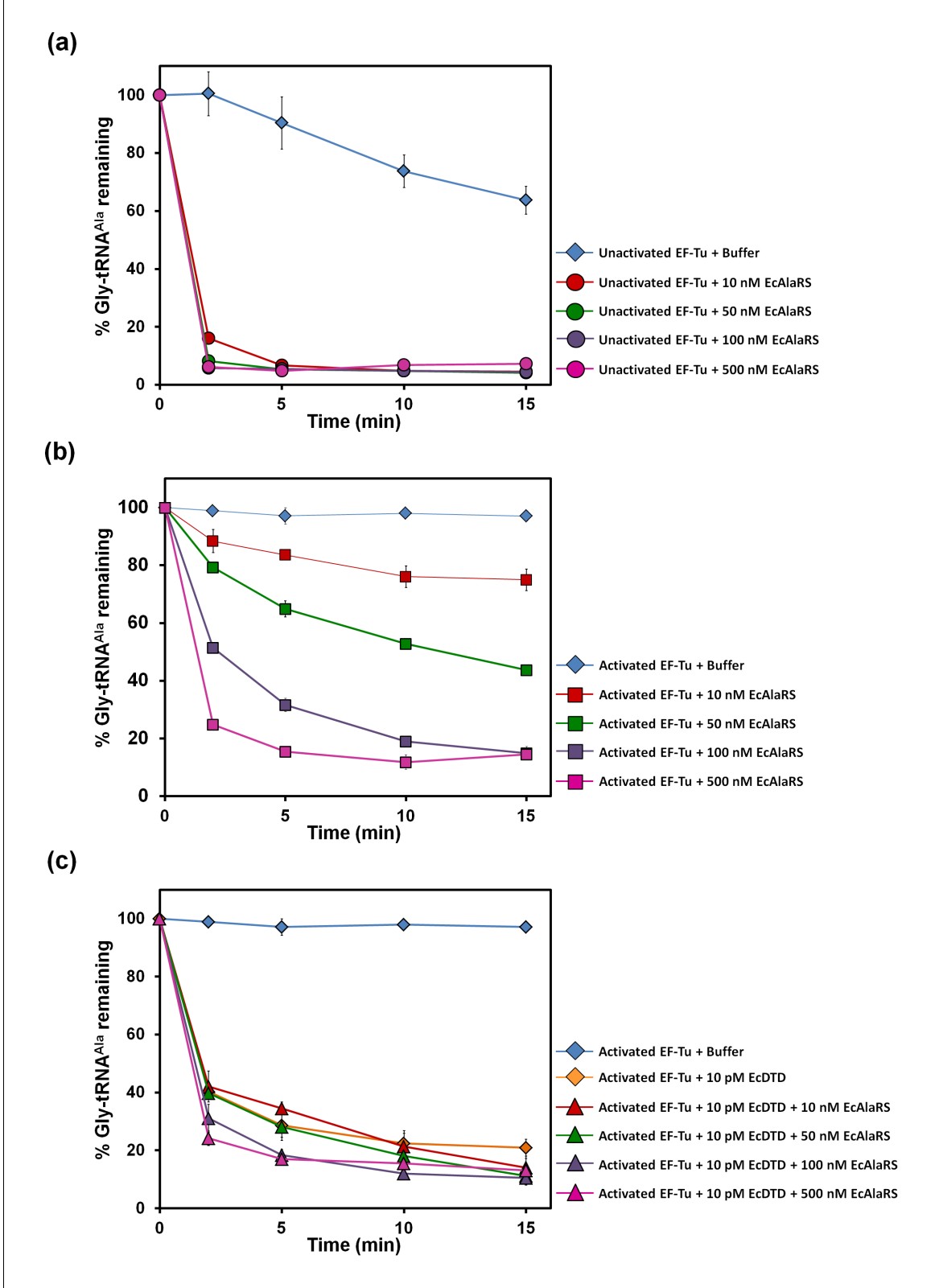

**Figure 3.** DTD has higher activity than AlaRS for the editing of Gly-tRNA[Ala]. (**a**) Deacylation of Gly-tRNA[Ala] in the presence of unactivated EF-Tu: buffer (blue diamond), 10 nM EcAlaRS (red circle), 50 nM EcAlaRS (green circle), 100 nM EcAlaRS (purple circle), 500 nM EcAlaRS (pink circle). (**b**) Deacylation of Gly-tRNA[Ala] in the presence of activated EF-Tu: buffer (blue diamond), 10 nM EcAlaRS (red square), 50 nM EcAlaRS (green square), 100 nM EcAlaRS (purple square), 500 nM EcAlaRS (pink square). (**c**) Deacylation of Gly-tRNA[Ala] by EcDTD and increasing concentration of EcAlaRS: buffer (blue

*Figure 3 continued on next page*

*Figure 3 continued*

diamond), 10 pM EcDTD (orange diamond), 10 pM EcDTD and 10 nM EcAlaRS (red triangle), 10 pM EcDTD and 50 nM EcAlaRS (green triangle), 10 pM EcDTD and 100 nM EcAlaRS (purple triangle), 10 pM EcDTD and 500 nM EcAlaRS (pink triangle). Error bars indicate one standard deviation from the mean of triplicate readings.

The following source data is available for figure 3:

**Source data 1.** Deacylation of Gly-tRNA$^{Ala}$ in the presence of unactivated and activated EF-Tu.

would have been so detrimental that a highly efficient factor like DTD was required to be employed for this function in addition to AlaRS editing domain.

## Editing-defective AlaRS and DTD knockout in *E. coli*

Earlier in vivo studies had shown that AlaRS editing defect causes glycine toxicity only at very high levels of glycine supplementation (80 mM) as opposed to serine which causes toxicity at significantly lower levels (2.5 mM) (*Beebe et al., 2003*). It is worth noting that these studies were carried out in strains harboring DTD, hence explaining the need for supplementation with more glycine to show toxicity. To check if the absence of DTD makes *E. coli* susceptible to glycine, we generated an *E. coli* strain in which *dtd* (the gene encoding DTD) was knocked out in the background of editing-defective AlaRS. To create a strain that was completely devoid of AlaRS editing activity, the genomic copy of AlaRS gene (*alaS*) was knocked out and a triple-mutant AlaRS (viz., T567F/S587W/C666F) was expressed from a plasmid. The editing site mutations were designed on the basis of a structural (homology) model of *E. coli* AlaRS *cis*-editing domain that was generated using the structure of *Archaeoglobus fulgidus* AlaRS (PDB id: 2ZTG) as a template. This model was then superimposed on *Pyrococcus horikoshii* AlaX complexed with serine (PDB id: 1WNU) (the best substrate-mimicking complex for AlaRS and AlaX available so far) (*Sokabe et al., 2005*). Three residues in the proposed editing site (*Beebe et al., 2003*; *Sokabe et al., 2005*) were supplanted by bulkier residues to occlude the pocket and prevent substrate binding (*Figure 4a,b*). The triple-mutant was found to be inactive on both Ser-tRNA$^{Ala}$ and Gly-tRNA$^{Ala}$ even when the protein concentration was increased to 1500-fold that of wild-type AlaRS (*Figure 4c,d*). It is worth mentioning here that the previously known editing-defective mutants of AlaRS (C666A and C666A/Q584H) (*Beebe et al., 2003*), when checked for deacylation activity on both Ser-tRNA$^{Ala}$ and Gly-tRNA$^{Ala}$, were found to show significant activity at just 10-fold higher concentration of the enzyme (*Figure 4c,d*). Thus, to completely abrogate AlaRS editing activity and to see the effect of editing from only DTD, we chose to use AlaRS triple-mutant for our cell-based toxicity studies.

## DTD prevents glycine toxicity in *E. coli*

Cellular toxicity studies using spot dilution assays and growth curve analysis with the DTD knockout strain in the background of AlaRS editing defect showed some toxicity even without any amino acid supplementation, and the toxicity increased with glycine supplementation as low as 3 mM. At 10 mM of glycine supplementation, the cells showed severe growth defect (*Figure 5a*). To check whether this was specifically due to mischarging caused by AlaRS, toxicity experiments were carried out in the presence of alanine, since the latter is expected to compete for the AlaRS aminoacylation site during charging of tRNA$^{Ala}$. It was observed that alanine supplementation completely recovered the toxicity caused by glycine and rescued the growth completely (*Figure 5b,c,d*). Moreover, to rule out any non-specific effects due to amino acid supplementation, histidine was used as a negative control, and it was found that it failed to rescue the cells from glycine toxicity (*Figure 5c,d*). Furthermore, DTD was found to be totally inactive on Ser-tRNA$^{Ala}$ in our biochemical assays which confirmed that the toxicity observed in the DTD-lacking cells in the background of AlaRS editing defect was not due to serine mischarging on tRNA$^{Ala}$ by AlaRS (*Figure 5—figure supplement 1*). The above observation is expected since it has been shown earlier that DTD's chiral proofreading site rejects even L-alanine, the smallest L-chiral substrate (*Routh et al., 2016*). Taken together, these experiments established that DTD acts as a key cellular factor that edits glycine mischarged on tRNA$^{Ala}$ by AlaRS. However, no toxicity of glycine supplementation was observed in *E. coli* MG1655 Δ*dtd* strain in the background of wild-type AlaRS (*Figure 5—figure supplement 2*). This indicates

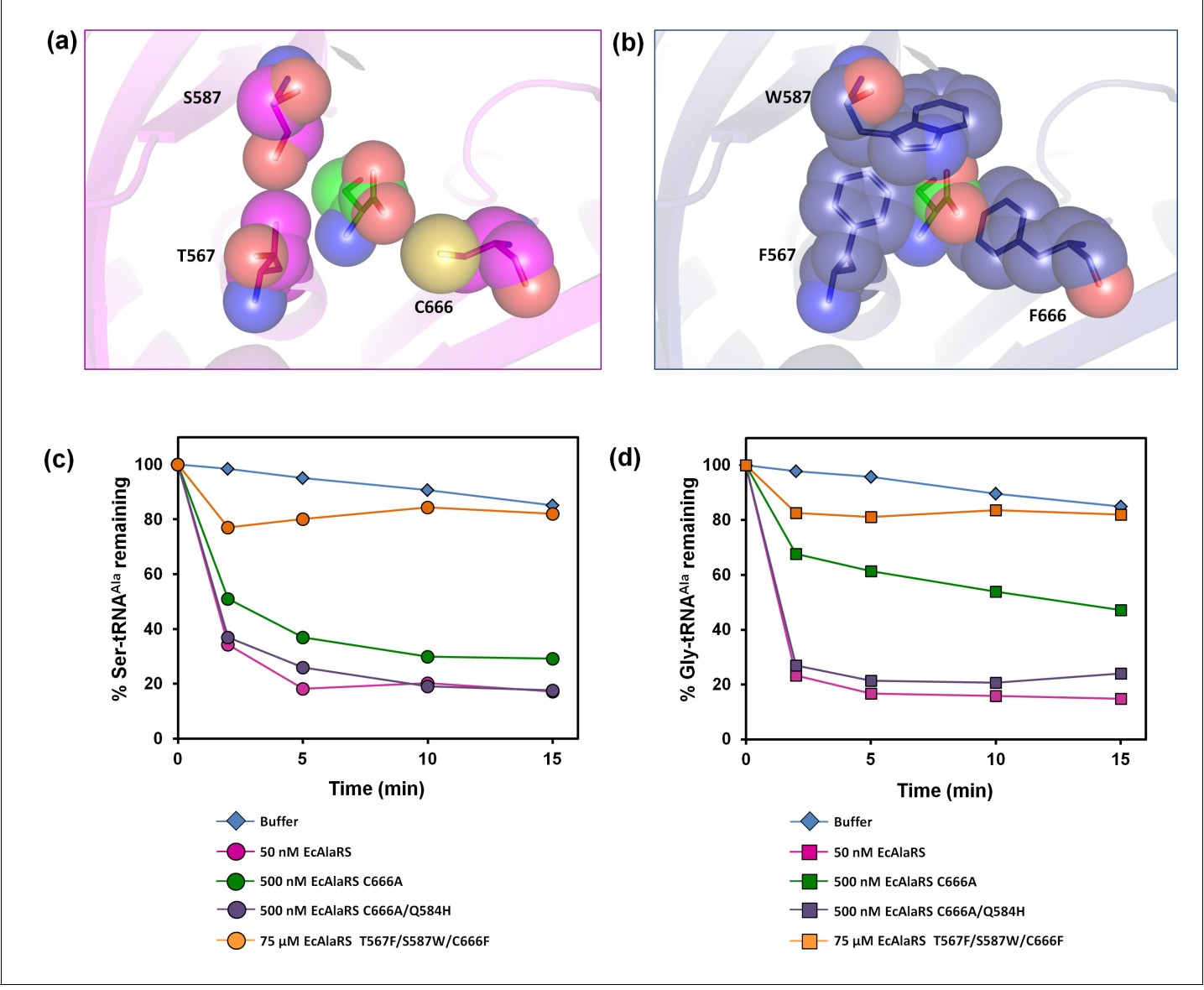

**Figure 4.** *E. coli* AlaRS editing site mutants. Homology model of *E. coli* AlaRS depicting serine (green sticks/spheres) in the editing site. *E. coli* AlaRS *cis*-editing domain was modeled using *A. fulgidus* AlaRS (PDB id: 2ZTG) as a template, whereas the position and orientation of serine in the model corresponds to that observed in serine-bound *P. horikoshii* AlaX structure (PDB id: 1WNU). (a) In the wild-type enzyme, residues selected for mutagenesis are represented as megenta sticks/spheres, showing an open pocket for substrate binding. (b) In AlaRS T567F/S587W/C666F, the mutated bulkier residues are depicted as blue sticks/spheres, showing occlusion of the pocket to prevent substrate binding. (c) Deacylation of Ser-tRNA$^{Ala}$ by buffer (blue diamond), 50 nM EcAlaRS (pink circle), 500 nM EcAlaRS C666A (green circle), 500 nM EcAlaRS C666A/Q584H (purple circle), 75 μM EcAlaRS T567F/S587W/C666F (orange circle). (d) Deacylation of Gly-tRNA$^{Ala}$ by buffer (blue diamond), 50 nM EcAlaRS (pink square), 500 nM EcAlaRS C666A (green square), 500 nM EcAlaRS C666A/Q584H (purple square), 75 μM EcAlaRS T567F/S587W/C666F (orange square).

The following source data is available for figure 4:

**Source data 1.** Deacylation of Ser-tRNA$^{Ala}$ and Gly-tRNA$^{Ala}$ by *E. coli* AlaRS editing site mutants.

that under normal growth conditions, AlaRS editing is sufficient for the cell to survive. The errors produced because of the absence of DTD are possibly tolerated by *E. coli* under laboratory conditions, as has been noted in several other cases of editing function of aaRSs (*Reynolds et al., 2010a*, *2010b*). The real implications of editing defects are only recently being appreciated in some specific

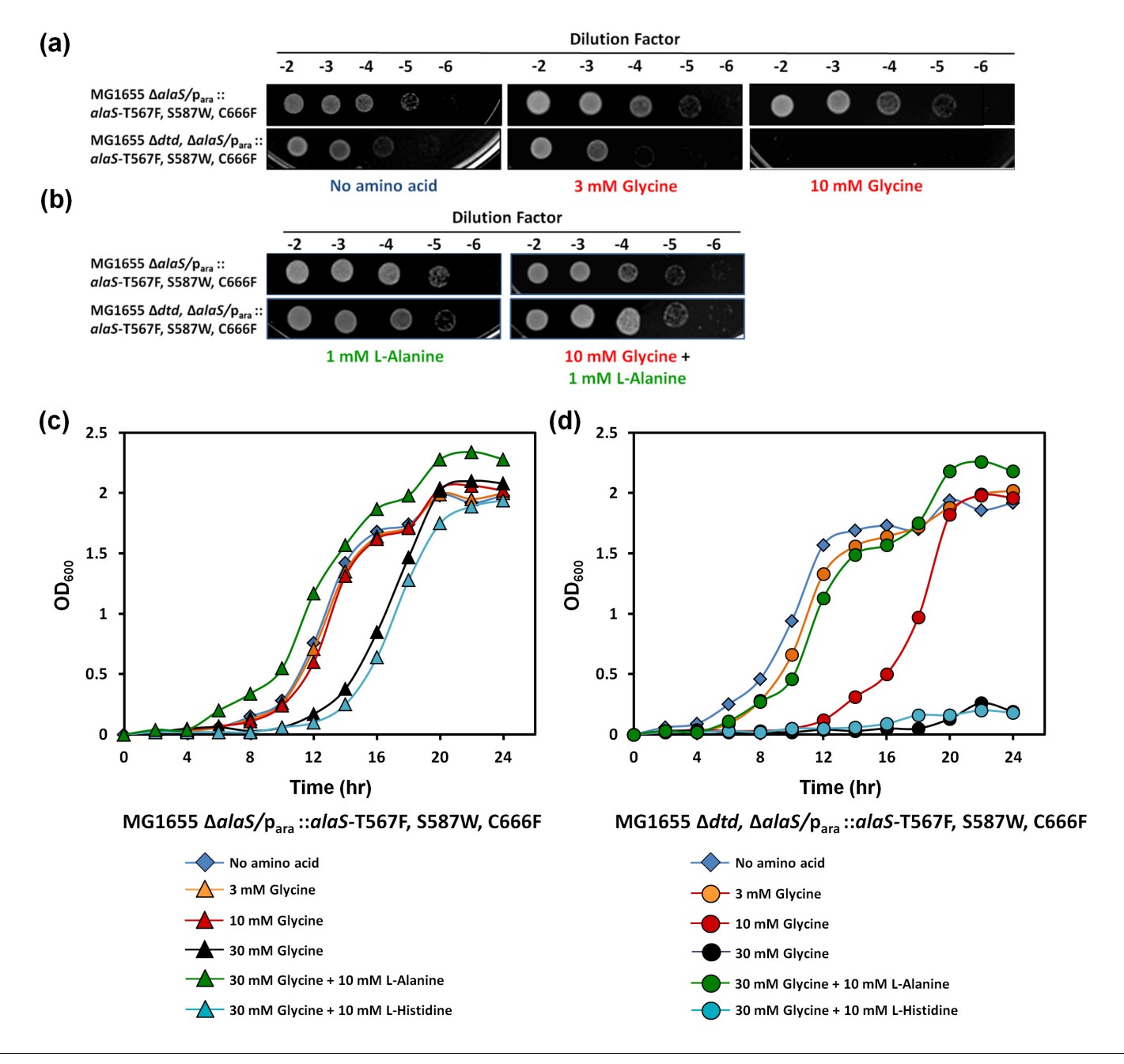

**Figure 5.** DTD knockout causes pronounced glycine toxicity in *E. coli*. Spot dilution assay of *E. coli* MG1655 Δ*alaS*/p$_{ara}$: : *alaS*-T567F, S587W, C666F compared with that of *E. coli* MG1655 Δ*dtd*, Δ*alaS*/p$_{ara}$:: *alaS*-T567F, S587W, C666F (**a**) in the presence of no amino acid, 3 mM glycine, or 10 mM glycine, and (**b**) in the presence of 1 mM L-alanine, or 10 mM glycine and 1 mM L-alanine. (**c**) Growth curve of *E. coli* MG1655 Δ*alaS*/p$_{ara}$: *alaS*-T567F, S587W, C666F supplemented with no amino acid (blue diamond), 3 mM glycine (orange triangle), 10 mM glycine (red triangle), 30 mM glycine (black triangle), 30 mM glycine and 10 mM L-alanine (green triangle), 30 mM glycine and 10 mM L-histidine (cyan triangle) (**d**) Growth curve of *E. coli* MG1655 Δ*dtd*, Δ*alaS*/p$_{ara}$: : *alaS*-T567F, S587W, C666F supplemented with no amino acid (blue diamond), 3 mM glycine (orange circle), 10 mM glycine (red circle), 30 mM glycine (black circle), 30 mM glycine and 10 mM L-alanine (green circle), 30 mM glycine and 10 mM L-histidine (cyan circle).

The following source data and figure supplements are available for figure 5:

**Source data 1.** Growth curves of *E. coli* MG1655 with and without *dtd* knockout in AlaRS editing-defective background.

**Figure supplement 1.** DTD is inactive on Ser-tRNA$^{Ala}$.

*Figure 5 continued on next page*

*Figure 5 continued*

**Figure supplement 2.** Spot dilution assay of *E. coli* MG1655 compared with *E. coli* MG1655 Δ*dtd* with increasing concentration of glycine.

growth conditions like oxidative stress, oxygen deprivation, starvation/nutrient limiting conditions etc. (*Bullwinkle et al., 2014*; *Cvetesic et al., 2014*; *Kermgard et al., 2017*).

## tRNA$^{Ala}$-specific G3•U70 wobble base pair acts as a positive determinant for DTD

Significantly high (~1000-fold higher) activity of DTD on Gly-tRNA$^{Ala}$ when compared to that on Gly-tRNA$^{Gly}$ (*Figure 6d*, *Figure 6—figure supplement 1*) indicated that tRNA$^{Ala}$ must have some positive determinants for DTD. Since G3•U70 is unique to and one of the major identity elements of tRNA$^{Ala}$ across all life forms (*Hou and Schimmel, 1988*; *1989*; *McClain and Foss, 1988*; *Ripmaster et al., 1995*; *Shiba et al., 1995*), we envisaged that DTD could be positively selecting tRNA$^{Ala}$ using the wobble base pair. To test this hypothesis, we transplanted G3•U70 at the same position in tRNA$^{Gly}$. We found that DTD had more than 10-fold increased activity on Gly-tRNA$^{Gly}$ harboring G3•U70 as compared to the wild-type achiral cognate substrate (*Figure 6a,c*). To strengthen this hypothesis, we substituted the G3•C70 in the wild-type Gly-tRNA$^{Gly}$ with the other Watson-Crick base pair, that is A3•U70. This substitution did not cause any increase in the activity of DTD (*Figure 6b*). This clearly suggests that the G3•U70 wobble base pair, which is a universally conserved feature of tRNA$^{Ala}$, acts as a positive determinant for DTD. It also suggests the existence of other features in tRNA$^{Ala}$ distinct from tRNA$^{Gly}$ that accounts for the higher activity of DTD on tRNA$^{Ala}$ when compared to tRNA$^{Gly}$, and this aspect requires further exploration. Since DTD is expected to act on all D-aminoacyl-tRNAs, it was assumed that there is no specificity code for its action on tRNAs. The identity determinant–switching experiment resulting in higher activity suggests for the first time an underlying tRNA-based code for DTD action.

## DTD is recruited throughout bacteria and eukaryotes for Gly-tRNA$^{Ala}$ removal

DTD's ubiquitous presence in bacteria and eukaryotes prompted us to test whether its role in clearing mischarged Gly-tRNA$^{Ala}$, in addition to its role in proofreading D-aminoacyl-tRNAs, is conserved in these two domains of life. It is important to investigate this aspect because we found significant differences in residues—which are believed to interact with the acceptor stem of tRNA—around the chiral proofreading site of DTD. To this end, we first superimposed crystal structures of DTD from various organisms and manually docked tRNA$^{Ala}$ (tRNA$^{Ala}$ was taken from PDB id: 3WQY) on the superposed structures. Since DTD is not expected to establish contacts beyond the acceptor stem of tRNA, we took into consideration only those residues of DTD that were within 6 Å from the 3′-terminal CCA-arm of tRNA (*Figure 7—figure supplement 1b*), and looked for their conservation/variation using structure-based multiple sequence alignment (*Figure 7—figure supplement 1a*). Notably, only 7 out of 20 residues in the selected region are invariant across bacterial and eukaryotic DTDs (*Figure 7—figure supplement 1*).

Given the significant differences observed in a region around the active site of DTD which is likely to interact with tRNA, it is not unreasonable to anticipate that they may affect DTD's activity on Gly-tRNA$^{Ala}$. To ascertain this, we tested DTDs from different eukaryotes—*Leishmania major* (LmDTD), *Drosophila melanogaster* (DmDTD) and *Danio rerio* (DrDTD)—spanning the entire spectrum of unicellular, invertebrate and vertebrate species, in addition to DTDs from *Plasmodium falciparum* (PfDTD) and *E. coli* (EcDTD). These DTDs were tested on glycine mischarged on both *E. coli* tRNA$^{Ala}$ and *D. melanogaster* tRNA$^{Ala}$ (*Figure 7*). All these DTDs were found to act effectively on both bacterial and eukaryotic tRNAs at 10 pM of DTD concentration. This indicates that in spite of the cross-species differences in DTD and tRNA$^{Ala}$, DTD's activity on Gly-tRNA$^{Ala}$ is most likely conserved throughout bacteria and eukaryotes.

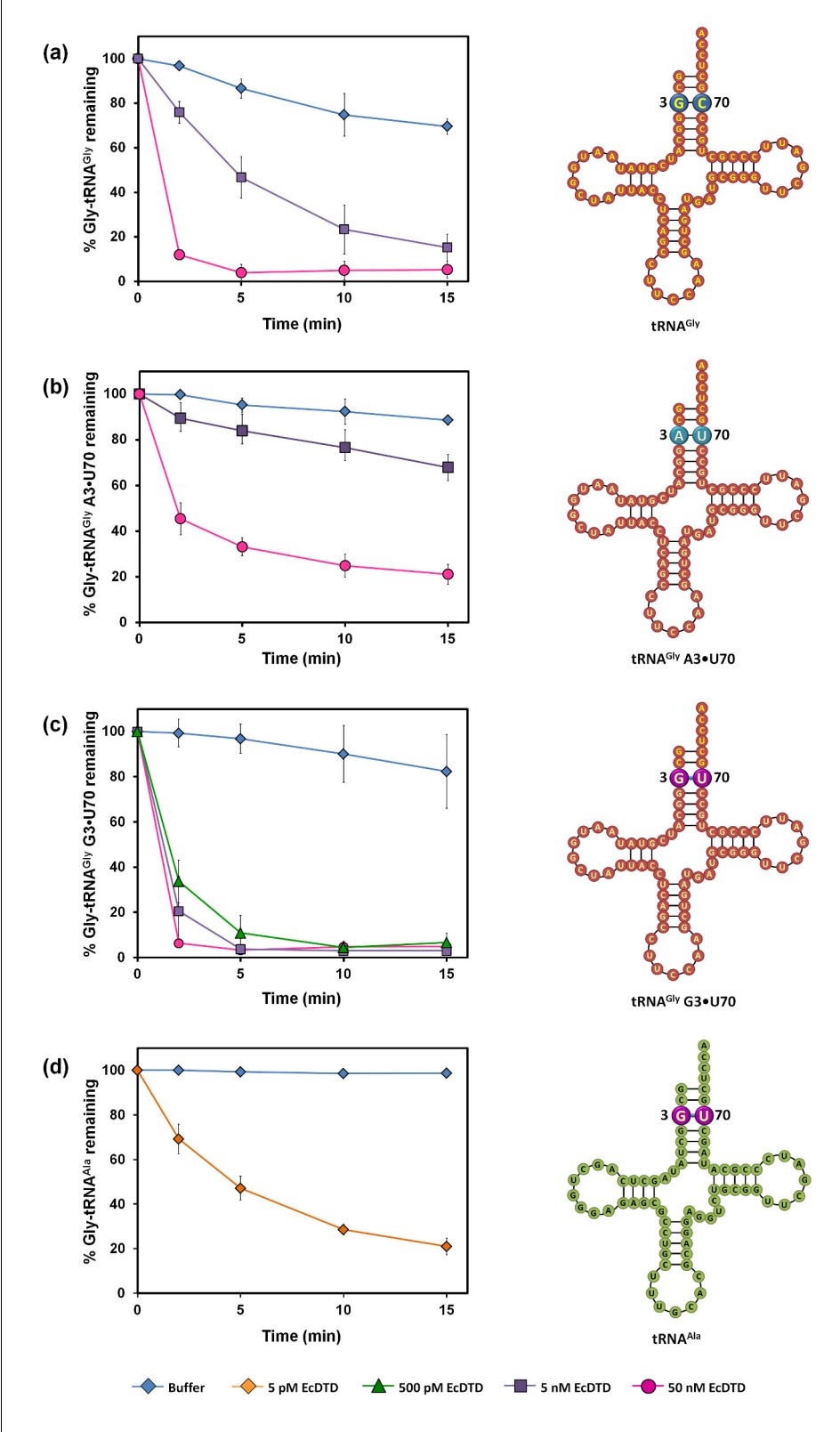

**Figure 6.** DTD positively selects the tRNA acceptor stem element G3•U70. (a) Deacylation of Gly-tRNA[Gly] by buffer (blue diamond), 5 nM EcDTD (purple square), 50 nM EcDTD (pink circle). (b) Deacylation of Gly-tRNA[Gly] A3•U70 by buffer (blue diamond), 5 nM EcDTD (purple square), 50 nM EcDTD (pink circle). (c) Deacylation of Gly-tRNA[Gly] G3•U70 by buffer (blue diamond), 500 pM EcDTD (green triangle), 5 nM EcDTD (purple square), 50 nM

*Figure 6 continued on next page*

*Figure 6 continued*

EcDTD (pink circle). (**d**) Deacylation of Gly-tRNA$^{Ala}$ by buffer (blue diamond), 5 pM EcDTD (orange diamond). Error bars indicate one standard deviation from the mean of triplicate readings.

The following source data and figure supplement are available for figure 6:

**Source data 1.** Deacylation of Gly-tRNA$^{Gly}$ mutants and Gly-tRNA$^{Ala}$ by DTD.

**Figure supplement 1.** DTD's activity on the cognate achiral substrate.

## Discussion

This study provides an unprecedented solution to a fundamental and long-standing puzzle by elucidating a hitherto unknown and physiologically important function of DTD, which was till now implicated only in enforcing homochirality during translation of the genetic code. The discovery of DTD as a key cellular factor for the elimination of Gly-tRNA$^{Ala}$ provides an elegant explanation as to why glycine mischarging by AlaRS was not encountered or considered as a cellular hazard in all the previous studies. So far, only serine mischarging on tRNA$^{Ala}$ by AlaRS was believed to be the major threat to the cell, since cells harboring editing-defective AlaRS would show toxicity only to low levels of serine but not glycine (*Beebe et al., 2003*; *Lee et al., 2006*). However, this observation seemed puzzling for two reasons. Firstly, glycine misactivation by AlaRS is known to occur at about twice the rate of serine misactivation (*Tsui and Fersht, 1981*). Secondly and more importantly, it is unlikely that a defect in the proofreading domain that edits both serine and glycine would cause toxicity only due to serine but not glycine. Moreover, for a protein's structure and function, the substitution of glycine for alanine is expectedly more subversive than the substitution of serine for alanine (*Betts et al., 2003*). Our study thus brings forth the criticality of glycine mischarging problem, which

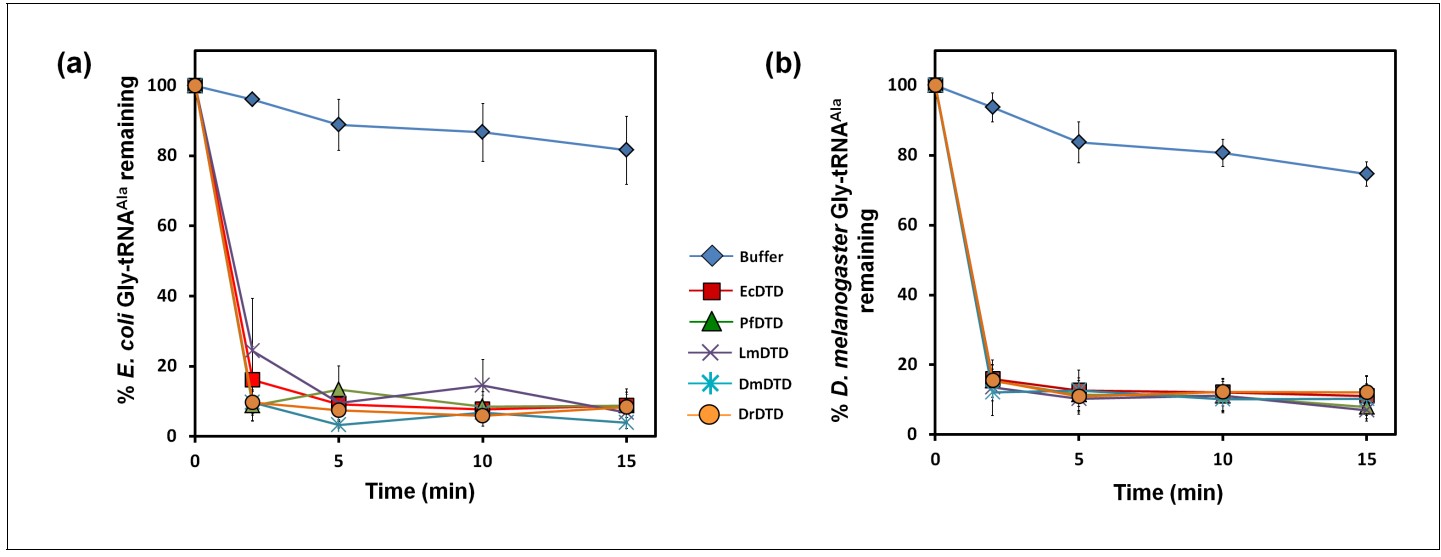

**Figure 7.** DTD edits Gly-tRNA$^{Ala}$ across bacteria and eukaryotes. (**a**) Deacylation of *E. coli* Gly-tRNA$^{Ala}$ by buffer (blue diamond), 10 pM EcDTD (red square), 10 pM PfDTD (green triangle), 10 pM LmDTD (purple cross), 10 pM DmDTD (cyan star), 10 pM DrDTD (orange circle). (**b**) Deacylation of *D. melanogaster* Gly-tRNA$^{Ala}$ by buffer (blue diamond), 10 pM EcDTD (red square), 10 pM PfDTD (green triangle), 10 pM LmDTD (purple cross), 10 pM DmDTD (cyan star), 10 pM DrDTD (orange circle). Error bars indicate one standard deviation from the mean of triplicate readings.

The following source data and figure supplement are available for figure 7:

**Source data 1.** Deacylation of *E. coli* Gly-tRNA$^{Ala}$ and *D. melanogaster* Gly-tRNA$^{Ala}$ by bacterial and eukaryotic DTDs.

**Figure supplement 1.** Variations in the tRNA-binding site of DTD.

was largely overlooked and underappreciated prior to this work. It also reveals that nature was forced to devise and retain throughout evolution a key checkpoint in the form of DTD that is more efficient than even AlaRS's proofreading function to tackle this problem (*Figure 8*). However, for *E. coli* under laboratory conditions, knockout of DTD in AlaRS editing-proficient background did not cause toxicity on glycine supplementation. Very likely, errors caused due to defect in proofreading by DTD knockout are tolerated by *E. coli*, as noted in several cases of other proofreading deficiencies in *E. coli* (*Reynolds et al., 2010a*, *2010b*). Moreover, defects in error correction are known to manifest in toxic effects only in special growth conditions like oxidative stress, oxygen deprivation, starvation etc. (*Bullwinkle et al., 2014*; *Cvetesic et al., 2014*; *Kermgard et al., 2017*).

The study also highlights the necessity of keeping DTD's active site design intact during the course of evolution, probably because removal of the mischarged Gly-tRNA$^{Ala}$ species from the cellular pool took precedence over DTD's unwarranted activity on Gly-tRNA$^{Gly}$. Nevertheless, the glycine 'misediting paradox' was effectively resolved by safeguarding the cognate achiral substrate using EF-Tu as well as keeping the cellular levels of DTD low and tightly regulated (*Routh et al., 2016*). Thus, what seemed to be an apparent flaw in the architecture of DTD's active site proved to be a necessity. Moreover, the dual activity of DTD on both achiral and D-chiral substrates depicts it as a plausible 'connecting link' or 'bridging factor' between D-chirality–based and the canonical L-chirality–based proofreading during protein biosynthesis (*Figure 8*). This view gains support from the fact that a DTD-like fold appended to archaeal threonyl-tRNA synthetase (ThrRS) as the N-terminal editing domain (NTD) is specific for editing L-serine mischarged on tRNA$^{Thr}$ (*Ahmad et al., 2015*; *Dwivedi et al., 2005*; *Hussain et al., 2006*, *2010*). It is also worth noting here that the DTD-like fold present in two functional contexts—as NTD in archaea, and as DTD in bacteria and eukaryotes—operates majorly through main chain-mediated contacts for substrate recognition and performs catalysis through RNA, suggesting its primordial origins (*Ahmad et al., 2013*, *2015*; *Routh et al., 2016*).

Glycine mischarging by AlaRS is inevitable and is a classic case of error made by the aminoacylation site of aaRS, whereas serine mischarging is an offshoot of amino group selection for alanine (*Guo et al., 2009*). DTD's significantly higher activity on Gly-tRNA$^{Ala}$ as compared to AlaRS suggests that wherever and whenever present, DTD plays the major role in clearing the non-cognate achiral substrate from the cellular pool. This probably helps the cell to overcome the double-discrimination problem that is encountered by AlaRS in all extant organisms. In the primordial scenario, DTD could

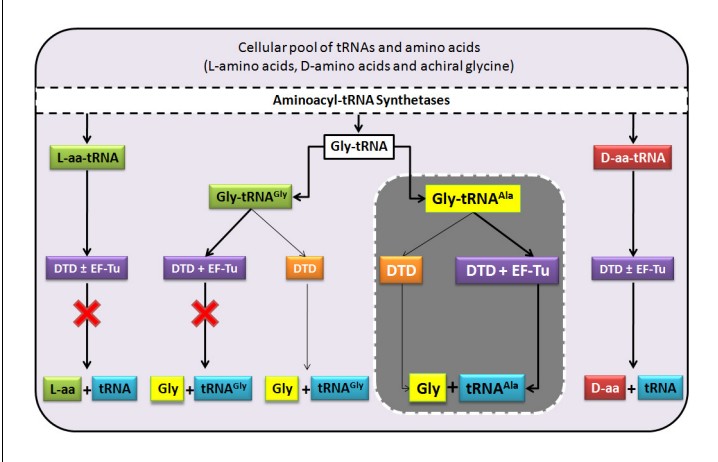

**Figure 8.** DTD doubles as a key factor to uncouple glycine mischarged on tRNA$^{Ala}$. In the cell, aminoacylation by aaRSs leads to the formation of different aa-tRNAs, of which L-aa-tRNAs (left extreme) are not acted upon by DTD, while D-aa-tRNAs (right extreme) are effectively decoupled in the presence or absence of EF-Tu, thereby enforcing homochirality. Glycylated tRNAs are acted upon by DTD (centre) but EF-Tu offers protection to the cognate Gly-tRNA$^{Gly}$ to prevent its misediting, while the mischarged/non-cognate Gly-tRNA$^{Ala}$ is efficiently cleared even in the presence of EF-Tu. Thick connecting arrows indicate the cellular scenario, wherein both DTD and EF-Tu are present.

have been primarily employed as a glycine-removing factor. This activity of DTD could have aided the formation of relatively rigid and stable peptide/protein scaffolds by precluding glycine misincorporation, since the latter would have been detrimental to their stability.

Another interesting facet of DTD that has emerged from this study is its ability to specifically recognize G3•U70 in the acceptor arm of tRNA$^{Ala}$. This wobble base pair is a unique and major identity determinant of tRNA$^{Ala}$ which marks it for both aminoacylation and deacylation by AlaRS from bacteria to humans (*Beebe et al., 2008*; *Hou and Schimmel, 1988*; *McClain and Foss, 1988*). The specificity of AlaRS for G3•U70 is so robust that incorporation of this base pair into other tRNAs or minihelices converts non-alanine-accepting tRNAs to be recognized and charged by AlaRS (*Musier-Forsyth and Schimmel, 1999*). This primordial mode of recognition was proposed to be an acceptor stem-based genetic code that could have been operational since the pre-tRNA era. The G3•U70-based selection of tRNA$^{Ala}$ by DTD clearly indicates towards the role of DTD in editing Gly-tRNA$^{Ala}$ even before the recruitment of AlaRS editing domains, which have primarily evolved to remove serine mischarged on tRNA$^{Ala}$ (*Novoa et al., 2015*). Furthermore, modulation of DTD's activity depending on tRNA elements is counter-intuitive as DTD is expected to act on multiple tRNAs with comparable efficiencies. Hence, the present work has unveiled a completely new aspect of DTD's aminoacyl-tRNA recognition code in which the role of the amino acid as well as the tRNA component needs to be looked at separately. Interestingly, a recognition code exists for EF-Tu, wherein three successive base pairs in the T-stem of tRNA thermodynamically compensate for the differential binding affinity of EF-Tu toward different amino acids (*LaRiviere et al., 2001*; *Roy et al., 2007*; *Sanderson and Uhlenbeck, 2007b*; *Schrader et al., 2009*). Thus, it becomes important to understand how DTD treats multiple aminoacyl-tRNAs using its own recognition code.

It is interesting to note that in certain cellular contexts, glycine as well as some D-amino acids can be present in relatively high concentrations, wherein these amino acids play important physiological roles. For example, in neuronal tissues, D-serine and D-aspartate along with glycine are abundant and act as neurotransmitters/neuromodulators (*Hashimoto and Oka, 1997*; *Snyder and Kim, 2000*). In such instances, especially in neuronal tissues, DTD's function and its corresponding up-regulation (*Zheng et al., 2009*) suggest an all-pervasive requirement of this protein from a primordial domain involved in perpetuation of homochirality to current-day proofreader in physiological context. It has been established that even a mild compromise in AlaRS editing for Ser-tRNA$^{Ala}$ causes severe pathological conditions, such as neurodegeneration and cardiomyopathy in mouse (*Lee et al., 2006*; *Liu et al., 2014*). In this regard, the role and regulation of DTD in various cellular contexts, and more importantly in higher eukaryotes, will be an important aspect which needs to be probed to gain newer insights into DTD's physiological significance.

## Materials and methods

### Cloning, expression and protein purification

DTD genes from the genome of *E. coli* and cDNAs of *P. falciparum*, *L. major*, *D. melanogaster* (fruit fly) and *D. rerio* (zebrafish) were cloned, and the proteins were expressed and purified as described previously (*Ahmad et al., 2013*; *Routh et al., 2016*). The gene (*alaS*) encoding *E. coli* AlaRS (EcAlaRS) cloned in pET-26b vector was a gift from Prof. William H. McClain (University of Wisconsin-Madison, USA). An N-terminal 6X His-tagged (N-His) construct was made for EcAlaRS in pET-28b using restriction-free cloning method (*van den Ent and Löwe, 2006*). The plasmid containing EcAlaRS gene was transformed in *E. coli* BL21(DE3) cells for protein overexpression. The N-His-tagged EcAlaRS protein was purified by a two-step protocol involving Ni-NTA affinity chromatography and size exclusion chromatography (SEC). The cells were lysed in lysis buffer containing 50 mM Tris–HCl pH 8.0, 150 mM NaCl, 5 mM 2-Mercaptoethanol ($\beta$-ME), and 10 mM imidazole. The same buffer was used to pre-equilibrate Ni-NTA column on which the cell lysate was loaded. After loading, the column was first washed with lysis buffer followed by wash buffer containing 50 mM Tris–HCl pH 8.0, 150 mM NaCl, 5 mM $\beta$-ME, and 30 mM imidazole. Protein was eluted with elution buffer containing 50 mM Tris–HCl pH 8.0, 150 mM NaCl, 5 mM $\beta$-ME, and 250 mM imidazole. The fractions containing protein of interest were pooled, concentrated and subjected to SEC purification using Superdex-200 in a buffer containing 100 mM Tris–HCl pH 8.0, 300 mM NaCl, and 10 mM $\beta$-ME. Finally, the

fractions containing purified protein were pooled, concentrated and mixed with equal volume of 100% glycerol before storing as aliquots at −30°C for further use.

The gene (*tufA*) encoding EF-Tu was PCR-amplified from the genomic DNA of *Thermus thermophilus*, and cloned in pET-28b using restriction-free cloning method (*van den Ent and Löwe, 2006*). The N-His-tagged EF-Tu protein from *T. thermophilus* was was then overexpressed in *E. coli* BL21 (DE3) cells and purified by a two-step method involving Ni-NTA affinity chromatography and SEC. The overall purification protocol remained very similar to the one described above except for changes in the buffer composition used in both steps. For Ni-NTA affinity chromatography, all the buffers contained 50 mM 4-(2-hydroxyethyl)-1-piperazineethanesulfonic acid (HEPES) titrated with potassium hydroxide (KOH), that is, HEPES-KOH, pH 7.5, 500 mM NaCl, 10 mM magnesium chloride ($MgCl_2$), 5% glycerol, 5 mM $\beta$-ME, and 100 µM guanosine-5′-diphosphate (GDP). Additionally, lysis, wash and elution buffers contained 10 mM, 30 mM and 250 mM imidazole, respectively. Following affinity chromatography, SEC was carried out using Sephadex G-200 and a buffer containing 50 mM HEPES-KOH pH 7.5, 500 mM ammonium chloride ($NH_4Cl$), 20 mM $MgCl_2$, 10% glycerol, 5 mM dithiothreitol (DTT), and 100 µM GDP. Finally, the purified protein was processed and stored at −30°C as described above. All protein purification steps from cell lysis onwards were carried out on ice or at 4°C.

The mutants were generated using QuickChange XL Site-directed kit (Stratagene, La Jolla, CA).

## Biochemical assays

*E. coli* tRNA$^{Ala}$ was charged with alanine, serine and glycine by EcAlaRS C666A mutant as described by *Pasman et al. (2011)*. The same protocol was followed to charge *D. melanogaster* tRNA$^{Ala}$ with glycine. *E. coli* tRNA$^{Gly}$ was charged with glycine by *T. thermophilus* GlyRS as described by *Routh et al. (2016)*. Deacylation assays with AlaRS and DTD were carried out as described by *Pasman et al. (2011)* and *Ahmad et al. (2013)*. EF-Tu activation was carried out as described by *Routh et al. (2016)*. It is to be noted that considering only 10–15% efficiency of EF-Tu activation reaction (*Cvetesic et al., 2013*; *Sanderson and Uhlenbeck, 2007a*), the effective (activated) EF-Tu concentration in our assay conditions was in the range of 200–300 nM when the total EF-Tu concentration used was 2 µM. Aminoacylation competition assays were performed in a solution of 100 mM HEPES pH 7.2, 2.5 mM DTT, 2 mM adenosine-5′-triphosphate (ATP), and 200 mM amino acid (alanine/serine/glycine) with 2 µM (total concentration) EF-Tu, 100 nM tRNA$^{Ala}$, 100 nM EcAlaRS and 10 pM EcDTD. These assays were performed at 37°C and were tracked for 15 min. Deacylation competition assays were also carried out in a solution of 100 mM HEPES pH 7.2, and 2.5 mM DTT with 2 µM (total concentration) EF-Tu, 200 nM aa-tRNA$^{Ala}$ and varying concentrations of EcAlaRS and EcDTD. Unless otherwise stated, the tRNAs used in the assays were from *E. coli*. Every data point denotes the mean of three independent readings. Error bars represent one standard deviation from the mean.

## Strain constructions

### Construction of Δ*alaS* deletion mutant

Wild-type *E. coli alaS* (WT) gene and editing-defective *alaS* triple-mutant T567F/S587W/C666F (TM) were cloned into *SacI*/*Hind*III sites of pBAD33 vector (Cam$^R$, pACYC origin) under the control of an arabinose-inducible promoter. *E. coli* MG1655 strain (RRID: SCR_002804) was transformed with pBAD33 WT-AlaRS and TM-AlaRS, and the transformants were selected on LB-agar plate containing chloramphenicol (20 µg ml$^{-1}$ final concentration) at 37°C. These strains were used for making Δ*alaS*::Kan deletion using P1 phage-mediated transductions (*Miller, 1992*). P1 phage for *alaS* knockout was prepared from a Keio collection deletion mutant JW 2667, which had a duplication of *alaS* (*Baba et al., 2006*). The presence of the deletion was confirmed by PCR amplification and sequencing the junctions of the deletion–insertion.

### Construction of Δ*dtd*Δ*alaS* deletion mutant

MG1655 Δ*dtd*::Kan was generated using P1 lysate from Keio collection (JW 3858–2). Marker-less Δ*dtd* was generated by flipping out the antibiotic resistance (Kan) marker by transforming with plasmid pCP20 (*Datsenko and Wanner, 2000*). This mutant strain was transformed with pBAD33 WT-AlaRS and TM-AlaRS. Δ*alaS*::Kan was introduced into these strains to generate Δ*dtd*Δ*alaS*. Deletion

mutants were selected on LB-agar plate containing kanamycin (25 μg ml$^{-1}$ final concentration), chloramphenicol (20 μg ml$^{-1}$ final concentration) and 0.4% (w/v) L-arabinose.

## Viability assays

Viability assays were performed with deletion mutant strains of $\Delta alaS$ and $\Delta dtd\Delta alaS$ in minimal medium (Miller, 1992). Relevant cultures were grown until OD$_{600}$ reached 0.6 and were 10-fold serially diluted ($10^{-2}$, $10^{-3}$, $10^{-4}$, $10^{-5}$ and $10^{-6}$). Of each serially diluted sample, 3 μl was spotted on minimal agar plates containing 0.002% L-arabinose, 0.2% maltose as carbon source, glycine (3 mM or 10 mM) and/or L-alanine (1 mM). The plates were incubated at 37°C for 20–36 hr.

For growth curves, primary cultures were grown in LB medium containing 0.0002% L-arabinose, Kanamycin and Chloramphenicol at 37°C until OD$_{600}$ reached 1.0. Subsequently, 2% inoculum was used to initiate 15 ml secondary culture in 1X minimal medium containing 0.2% maltose as carbon source and 0.0002% L-arabinose. The secondary culture was grown at 37°C to obtain a cell density (OD$_{600}$) of ~0.6. These cultures were again grown in 1X minimal medium with/without amino acids (glycine, L-alanine and L-histidine) of the indicated concentrations. The growth was monitored at every 2-hr interval. All the experiments were done in triplicates.

## Acknowledgements

The authors acknowledge Dr. Manjula Reddy and L. Sai Sree for their help with cell-based toxicity assays. We thank NBRP (Japan): *E. coli* for Keio collection mutants. KIP and SBR thank Council of Scientific and Industrial Research (CSIR), India, KS thanks DBT-RA Programme, India, and SKK thanks DST-INSPIRE, India, for research fellowships. RS acknowledges funding from 12th Five Year Plan Project BSC0113 of CSIR, India, JC Bose Fellowship of SERB, India and Centre of Excellence Project, DBT, India. The funding agencies had no role in study design, analysis, decision to publish or preparation of the manuscript.

## Additional information

### Funding

| Funder | Grant reference number | Author |
| --- | --- | --- |
| Council of Scientific and Industrial Research | 12th Five Year Plan Project BSC0113 | Rajan Sankaranarayanan |
| Science and Engineering Research Board | JC Bose Fellowship | Rajan Sankaranarayanan |
| Department of Biotechnology, Ministry of Science and Technology | Centre of Excellence | Rajan Sankaranarayanan |
| Council of Scientific and Industrial Research | Research Fellowship | Komal Ishwar Pawar Satya Brata Routh |
| Department of Biotechnology, Ministry of Science and Technology | Research Associateship | Katta Suma |
| Department of Biotechnology, Ministry of Science and Technology | INSPIRE Fellowship | Santosh Kumar Kuncha |

The funders had no role in study design, data collection and interpretation, or the decision to submit the work for publication.

### Author contributions

KIP, Conceptualization, Data curation, Formal analysis, Investigation, Methodology, Writing—original draft, Writing—review and editing; KS, SPK, Data curation, Formal analysis, Investigation, Methodology, Writing—review and editing; AS, Data curation, Formal analysis, Investigation, Methodology; SKK, Data curation, Investigation, Methodology; SBR, Investigation, Methodology, Writing—review

and editing; RS, Conceptualization, Resources, Supervision, Funding acquisition, Validation, Writing—original draft, Project administration, Writing—review and editing

**Author ORCIDs**
Komal Ishwar Pawar, http://orcid.org/0000-0002-1968-9851
Rajan Sankaranarayanan, http://orcid.org/0000-0003-4524-9953

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
