## [Decision Letter]

Thank you for submitting your article "DTD's role beyond chiral proofreading as a cellular defense against glycine mischarging by AlaRS" for consideration by *eLife*. Your article has been reviewed by three peer reviewers, one of whom, Jonathan P Staley (Reviewer #1), is a member of our Board of Reviewing Editors, and the evaluation has been overseen by and Michael Marletta as the Senior Editor. The following individuals involved in review of your submission have agreed to reveal their identity: Michael Ibba (Reviewer #3).

The reviewers have discussed the reviews with one another and the Reviewing Editor has drafted this decision to help you prepare a revised submission

Summary:

tRNA charging fidelity is a fundamental issue in promoting protein synthesis and cellular fitness. In this manuscript, the authors define a novel mechanism for editing Gly-tRNA(Ala) by the universal factor DTD, previously known to function only in discrimination against tRNAs mischarged with D-amino acids. While editing Ser-tRNA(Ala) has been shown to be physiologically important, evidence that editing of Gly-tRNA(Ala) has not, even though tRNA(Ala) is as efficiently mischarged with Gly as with Ser, as the authors show. The authors further show that the canonical editing activity of AlaRS is inefficient in cleaving Gly-tRNA(Ala) but that DTD does cleave efficiently – many orders of magnitude more efficiently. While EF-Tu protects Gly-tRNA(Gly) from DTD, EF-Tu does not protect Gly-tRNA(Ala), and DTD acts orders of magnitude more efficiently on Gly-tRNA(Ala) than Gly-tRNA(Gly), highlighting the specificity of DTD for Gly-mischarged tRNA. Strikingly, the authors show that a knockout of DTD in *E. coli* (in an editing-deficient AlaRS background) renders concentrations of glycine toxic that can be rescued by alanine, establishing both the physiological challenge of glycine mischarging and the role of DTD in editing. The authors further show that this activity is conserved across all branches of life and unexpectedly appears to require recognition of the tRNA body. Overall, this elegant and clever manuscript resolves a long-standing question in the field of translation quality control.

Essential revisions:

1) A main question is whether the role of DTD in editing Gly-tRNA(Ala) is greater than the role of AlaRS. The physiological role for DTD was determined in a background deficient for editing by AlaRS. If the role of DTD is critical, it's role should remain important in a wild-type AlaRS background, which the authors should test. Either possible outcome will be informative.

2) Table 1: kcat/Km values are much higher than the typical diffusion-limited, macromolecular reaction at ~100 uM-1/s^-1^ (table 1 values are >10,000 uM-1/s^-1^). This does not seem to be possible. How can these values be justified? There may be some fundamental issues in these measurements.

3) Subsection “DTD effectively decouples glycine mischarged on tRNA^Ala^” and Figure 3: It's unclear how the conclusion that ~20,000 times more AlaRS than DTD is needed for Gly-tRNAAla deacylation is derived from Figure 3 data. I assume that the authors are comparing Figure 3 without an active EF-Tu? Or is it a comparison of 3B with 2B? In order to make this strong statement, one needs to at least take earlier time points than 2 min, and not in mixed AlaRS/DTD experiment where these two proteins may interact; indeed, because one cannot be sure which protein is responsible for the activity in 3C and because 3C is somewhat redundant with Figure 2 and Figure 3, the panel 3C should be removed. In addition, why does the DTD only deacylate ~75% of misacylated tRNA (3C), whereas AlaRS deacylates almost all tRNA (3A)?

4) Figure 6: the requirement of G3U70 for DTD deacylation is a major conclusion of the paper. Yet, this conclusion is based entirely on one single mutation in tRNAGly. This figure is so poorly presented that one cannot even see what the sequence differences are between the *E. coli* tRNAGly and Ala used. More mutation data, e.g. from other tRNAGly isoacceptors are needed. It is also crucial to mutate tRNAAla G3U70 to show a severe reduction of the deacylation activity. To bypass the difficulty of making sufficient amount of Gly-tRNAAla G3U70 mutant by AlaRS, one could use flexizyme to make these substrates. Without such experiments, the authors would need to dramatically tone down the significance and implications of this result concerning a single mutation of a single tRNA.

5) The conclusions are overstated for the experiments to determine how applicable this mechanism is across different domains (subsection “DTD is recruited throughout bacteria and eukaryotes for Gly-tRNAAla removal”). While the results shown are certainly interesting, the authors should be a little more circumspect.

---

## [Author Response]

*Essential revisions:*

*1) A main question is whether the role of DTD in editing Gly-tRNA(Ala) is greater than the role of AlaRS. The physiological role for DTD was determined in a background deficient for editing by AlaRS. If the role of DTD is critical, it's role should remain important in a wild-type AlaRS background, which the authors should test. Either possible outcome will be informative.*

We thank the reviewers for pointing out an important aspect of the effect of DTD knockout in the background of wild-type AlaRS. We have now done the experiments to check the effect of glycine supplementation on both *E. coli* MG1655 and *E. coli* MG1655 ∆*dtd* strains. We do not observe glycine toxicity in *E. coli* MG1655 ∆*dtd* strain in the background of wild-type AlaRS. This suggests that for *E. coli*, in the absence of DTD, AlaRS editing is sufficient to prevent glycine toxicity under laboratory conditions. This could be due to AlaRS’s ability to recycle Gly-tRNA^Ala^ in the absence of DTD and/or errors produced because of the absence of DTD are tolerated by *E. coli* under laboratory conditions.

Although translational quality control functions are known to be conserved in all domains of life, it is very well known that they are not essential for cellular growth under most laboratory conditions (Reynolds *et al.*, 2010a). The real implications of editing defects are only recently being appreciated in some specific growth conditions like oxidative stress, oxygen deprivation, starvation or nutrient-limiting conditions, etc. (Bullwinkle *et al.*, 2014; Cvetesic *et al.*, 2014; Kermgard *et al.*, 2017). These editing defects are also found to show differential effects in different cell compartments and cell types (Reynolds *et al.*, 2010b; Liu *et al.*, 2014).

In many cases, the importance of editing functions is found only in slow growth conditions. During exponential growth of both *E. coli* and yeast, certain translational quality control processes are also found to be dispensable (Reynolds *et al.*, 2010a; Reynolds *et al.*, 2010b). To test the contribution of only DTD in proofreading Gly-tRNA^Ala^, we carried out our experiments in the background of AlaRS editing defect (the so far known editor of Gly-tRNA^Ala^). As correctly pointed out by the reviewers, we have now included the data comprising toxicity assays of *E. coli* MG1655 ∆*dtd* strain in wild-type AlaRS background in the manuscript (Figure 5—figure supplement 2) and have also added a few sentences regarding this effect in the main text.

*2) Table 1: kcat/Km values are much higher than the typical diffusion-limited, macromolecular reaction at ~100 uM-1/s^-1^ (table 1 values are >10,000 uM-1/s^-1^). This does not seem to be possible. How can these values be justified? There may be some fundamental issues in these measurements.*

We are grateful to the reviewers for highlighting a very important aspect of measuring the kinetic parameters for DTD. As rightly pointed out by the reviewers, we find the k_cat_/K_m_ values to be significantly higher than the typical diffusion-limited macromolecular reaction. In this case, we have found that our K_m_ values are not very different from those estimated for several other editing modules (Pasman *et al.*, 2011), but our k_cat_ values are too high, thus tilting the overall enzyme efficiency (k_cat_/K_m_). One of the major reasons for this is that DTD is active at low picomolar concentrations, whereas other editing modules work at best only in nanomolar concentration range.

We have also discussed this issue of unusual values of kinetic parameters of DTD with some of the experts in enzyme kinetics within the country and outside but have still not been able to figure out the exact reasons for such high k_cat_/K_m_ values. There are a few enzymes, like 4-oxalocrotonase tautomerase and superoxide dismutase, which have been reported to cross the diffusion limits (Traut, 2008), but to come to a conclusion that DTD comes in that category, we need to establish these numbers thoroughly using different methods. A couple of issues that could have led to an aberrant estimation are given below:

a. Error in protein concentration estimation.

b. Error in estimation of initial velocities, since DTD is a fast enzyme.

To rule out the possibility that we underestimated the enzyme concentration, we have again measured the concentration of DTD and AlaRS, with bovine serum albumin (BSA) as a standard protein, using multiple methods (Absorbance at 280 nM (A_280_), Bradford assay and Bicinchoninic acid assay) which work on different principles for estimation of protein concentration. We have found that the estimations using these methods are similar to our earlier estimations (Table 1) which were carried out using A_280_.

Author response table 1.Protein concentration estimation.**DOI:**
http://dx.doi.org/10.7554/eLife.24001.022**Protein Name****Concentration of protein taken (mg/ml) measured using A_280_****Protein concentration estimation (mg/ml) using****Bicinchoninic acid Assay****Bradford Assay*****E. coli* AlaRS****4****3.82****5.22****8****7.45****9.02*****E. coli* DTD****4****4.36****4.83****8****8.08****7.93**

Even after further validation if the kinetic parameters are found to be similar and high as represented here, it will have serious impact and implication on enzyme kinetics in general and translational proofreading in particular, and one would also need to explain this in the context of physiological relevance and evolutionary implications. This would include generating slower variants and then to test their impact on rescuing DTD-induced phenotype.

Nevertheless, since the focus of the manuscript is that DTD functions to edit Gly-tRNA^Ala^, we felt that the kinetics data as represented in Table 1 (in the manuscript) do not directly contribute to the overall implications of the work. The kinetic constants of DTD for different substrates have to be looked into as a whole using multiple methodologies before presenting such a high value, as also noted correctly by the reviewers. Therefore, we propose to remove this table and once again thank the reviewers for pointing out to us the abnormal values (able 1 has been removed from the revised manuscript). However, as suggested by them in the next concern, the relative editing efficiencies of AlaRS and DTD are to be compared, which we have addressed below.

*3) Subsection “DTD effectively decouples glycine mischarged on tRNA^Ala^” and Figure 3: It's unclear how the conclusion that ~20,000 times more AlaRS than DTD is needed for Gly-tRNAAla deacylation is derived from Figure 3 data. I assume that the authors are comparing Figure 3 without an active EF-Tu? Or is it a comparison of 3B with 2B? In order to make this strong statement, one needs to at least take earlier time points than 2 min, and not in mixed AlaRS/DTD experiment where these two proteins may interact; indeed, because one cannot be sure which protein is responsible for the activity in 3C and because 3C is somewhat redundant with Figure 2 and Figure 3, the panel 3C should be removed. In addition, why does the DTD only deacylate ~75% of misacylated tRNA (3c), whereas AlaRS deacylates almost all tRNA (3A)?*

We thank the reviewers for bringing forth this point and we regret that our projection has inadvertently led to some confusion while reading the manuscript. The comparison we want to make is between Figure 2 and Figure 3 in the manuscript, where in the presence of activated EF-Tu, DTD at 5 pM concentration and AlaRS at 100 nM concentration show similar activities on Gly-tRNA^Ala^ under identical reaction conditions, thus leading to a difference of ~20,000-folds between the two enzymes. Also, for this comparison, we have not used a mixture of enzymes. DTD and AlaRS were used in separate reactions for checking their activities in the presence of activated EF-Tu (Figure 2 and Figure 3, respectively, in the manuscript). We have now mentioned explicitly in the revised manuscript that we are comparing Figure 2 and Figure 3, so that the readers will not have this confusion, as in the original manuscript.

In Figure 3 in the manuscript, we are checking the activity of DTD on Gly-tRNA^Ala^ in the absence or presence of increasing concentration of AlaRS. We find that the presence or absence of AlaRS does not make much difference in the deacylation of Gly-tRNA^Ala^ by DTD. Figure 3 is different from other experiments as this is the only competition deacylation experiment between DTD and AlaRS to deacylate Gly-tRNA^Ala^. We would like to keep this figure (retained in the revised manuscript) as now the original confusion between the earlier figures is resolved. However, if the reviewers still feel that this dataset is redundant and unnecessary, we can remove it from the manuscript.

Figure 3 in the manuscript shows deacylations of Gly-tRNA^Ala^ in the presence of unactivated EF-Tu. In this case, unactivated EF-Tu does not offer any protection to the substrate and thus, in general, higher deacylations are observed (as also seen for the buffer/no enzyme control). Figure 3 in the manuscript are deacylations of Gly-tRNA^Ala^ in the presence of activated EF-Tu. Here, activated EF-Tu offers some protection to the substrate (as seen by negligible deacylation in buffer/no enzyme control) and in both the reactions, AlaRS and DTD deacylate ~75% of the misacylated tRNA (Figure 3, respectively).

4) Figure 6: the requirement of G3U70 for DTD deacylation is a major conclusion of the paper. Yet, this conclusion is based entirely on one single mutation in tRNAGly. This figure is so poorly presented that one cannot even see what the sequence differences are between the E. coli tRNAGly and Ala used. More mutation data, e.g. from other tRNAGly isoacceptors are needed. It is also crucial to mutate tRNAAla G3U70 to show a severe reduction of the deacylation activity. To bypass the difficulty of making sufficient amount of Gly-tRNAAla G3U70 mutant by AlaRS, one could use flexizyme to make these substrates. Without such experiments, the authors would need to dramatically tone down the significance and implications of this result concerning a single mutation of a single tRNA.

We agree with the reviewers that the *E. coli* tRNA^Gly^ and *E. coli* tRNA^Ala^ figures should have details about their sequence. As correctly suggested by the reviewers, we have now included the sequence details of the tRNAs used in the study (in the improved version of the figure). Regarding DTD’s role in positively selecting the G3•U70 wobble base pair on tRNA, we have now performed additional experiments following the reviewers’ advice to further strengthen our claim. We have carried out deacylation experiments with *E. coli* Gly-tRNA^Gly^ A3•U70 and found that this tRNA variant does not show increase in activity compared to wild-type *E. coli* Gly-tRNA^Gly^ substrate (Figure 6). This indicates that DTD positively selects the structure and/or chemical interaction coming especially from G3•U70 wobble base pair. Besides, as per the suggestion of the reviewers, we have compared the three isoacceptors of tRNA^Gly^ and have found that their acceptor stem is identical (Figure 9). Since DTD is not expected to interact beyond the acceptor stem of tRNA, we do not anticipate the plausibility of variation in DTD’s activity on different isoacceptors of tRNA^Gly^.

We also agree with the reviewers that it is crucial to mutate G3•U70 of tRNA^Ala^ to show reduction of deacylation activity. In this regard, we have generated an aminoacylation site mutant of *E. coli* AlaRS, *viz.*, D400A. The corresponding mutation in AlaRS from *Archaeoglobus fulgidus* (D450A) has been previously shown to aminoacylate tRNA^Ala^ A3•U70 (Naganuma *et al.,* 2014). However, using *E. coli* AlaRS D400A mutant, we were not able to generate sufficient amounts of Gly-tRNA^Ala^ A3•U70 substrate to carry out deacylation reactions. Another strategy, as rightly indicated by the reviewers, is to use flexizymes to generate the desired substrate. We need to establish the system to test the above in the future.

Author response image 1.*E. coli* tRNA^Gly^ isoacceptors showing identical acceptor stem.**DOI:**
http://dx.doi.org/10.7554/eLife.24001.023

5) The conclusions are overstated for the experiments to determine how applicable this mechanism is across different domains (subsection “DTD is recruited throughout bacteria and eukaryotes for Gly-tRNAAla removal”). While the results shown are certainly interesting, the authors should be a little more circumspect.

We have now made the necessary changes in the manuscript to tone down the section as per the suggestion of the reviewers.